# Increased Proximal Wall Shear Stress of Basilar Artery Plaques Associated with Ruptured Fibrous Cap

**DOI:** 10.3390/brainsci12101397

**Published:** 2022-10-17

**Authors:** Ruiyun Huang, Hongbing Chen, Chenghao Li, Chaowei Lie, Zhihua Qiu, Yongjun Jiang

**Affiliations:** 1Department of Neurology, The Second Affiliated Hospital of Guangzhou Medical University, 250 Changgang East Road, Guangzhou 510260, China; 2Department of Neurology and Stroke Center, The First Affiliated Hospital, Sun Yat-Sen University, 58 Zhongshan Road II, Guangzhou 510080, China; 3Department of Radiology, The Second Affiliated Hospital of Guangzhou Medical University, 250 Changgang East Road, Guangzhou 510260, China

**Keywords:** rupture of fibrous cap, basilar artery, high resolution MRI, computational fluid dynamics, wall shear stress

## Abstract

Plaque rupture of the basilar artery is one of the leading causes of posterior circulation stroke. The present study aimed to investigate the role of fluid dynamics in the ruptured fibrous cap of basilar artery plaques. Patients with basilar artery plaques (50–99% stenosis) were screened. Integrity of the fibrous cap was assessed by high-resolution MRI. Computational fluid dynamics models were built based on MR angiography to obtain the wall shear stress and velocity. A total of 176 patients were included. High-resolution MRI identified 35 ruptured fibrous caps of basilar artery plaques. Ruptured fibrous cap was significantly associated with acute infarction (27/35 vs. 96/141, *p* < 0.05) in the territory of the basilar artery. Proximal wall shear stress of stenosis was positively related with the ruptured fibrous cap (OR 1.564; 95% CI, 1.101–2.222; *p* = 0.013). The threshold of wall shear stress for the ruptured fibrous cap of basilar artery plaques was 4.84 Pa (Area under ROC 0.732, *p* = 0.008, 95%CI 0.565–0.899). The present study demonstrated that increased proximal wall shear stress of stenosis was associated with ruptured fibrous caps of basilar artery plaques.

## 1. Introduction

Plaque rupture with superimposed thrombosis is one of the leading causes of ischemic stroke [1]. The main characteristic of vulnerable plaque is a large lipid core underneath a thin fibrous cap. When the fibrous cap is broken, platelets adhere to the surface of the ruptured fibrous cap (RFC) and form the subsequent thrombus [2,3]. Thus, it was necessary to discover why the plaques were broken. 

A principle is that the plaques would be broken when the external stress exceeds the material strength of a fibrous cap [4]. Wall shear stress (WSS), defined as the force per unit area exerted by the wall on the fluid in a direction on the local tangent plane, is the major external stress on the plaque surface. An increased WSS accelerates the atherogenic process [5,6] and is associated with embolism during carotid artery endarterectomy [7]. For the intracranial artery, increased WSS promoted aneurysm ruptures [8]. Leng et al. showed that a high WSS for the intracranial artery predicted stroke onset within one year in the same area [9]. There were some patients with basilar artery (BA) stenosis in their study. However, no report had investigated the role of WSS in the RFC. In the present study, we aimed to investigate whether WSS was an independent risk factor for RFC of BA plaques.

## 2. Materials and Methods

### 2.1. Patient Profile

The research protocol was reviewed and approved by the ethics committees of each institute according to the principles expressed in the Declaration of Helsinki. Written consent was obtained from the patients or their authorized relatives before enrollment.

Patients at the centers (Guangzhou Medical University and Sun Yat-Sen University) who had a stroke were screened from 1 June 2014, to 30 June 2019. The patients were enrolled if they met the following inclusion criteria: (1) older than 18 years old; (2) MRI (including T1-, T2-, and diffusion-weighted imaging, DWI) and magnetic resonance angiography (MRA); (3) BA atherosclerotic plaque (50–99% stenosis); and (4) successful HR-MRI. Patients were excluded if they met any of the following criteria: (1) missing clinical or imaging information; (2) cardioembolism; (3) MRA did not cover the whole BA; (4) occlusion of the BA; (5) failure of the computational fluid dynamics (CFD) calculation; (6) nonatherosclerotic stenosis, such as dissection, aneurysm, Moyamoya disease, or BA fenestration; (7) occlusion of the bilateral intracranial vertebral artery; or (8) no visualization of the fibrous cap. If the intracranial vertebral arteries terminated in the posterior inferior cerebellar artery, which is referred to as “basilarization” of one ICVA by Louis Caplan [10], or if one of the ICVAs were occluded, the origin of BA was considered the medullary-pontine junction. The stroke onset was defined as an infarction located in the territory of the posterior circulation except for the medulla in the DWI images.

### 2.2. MRI and High Resolution MRI (HR-MRI) Scanning

All the patients underwent MRI on admission using a 3.0 T MRI unit (GE or Siemens). The scanning sequences included T1-weighted imaging, T2-weighted imaging, fluid-attenuated inversion recovery sequence, DWI, apparent diffusion coefficient maps, and time-of-flight MRA (TOF-MRA) covering the Circle of Willis.

The HR-MRI protocol consisted of T1-weighted imaging with contrast injection (repetition time [TR], 600 ms; echo time [TE], 12 ms; field of view [FOV], 12 × 12 cm; matrix size, 384 × 219; number of excitations [NEX], 4), T2-weighted imaging (TR, 2910 ms; TE, 70 ms; FOV, 12 × 12 cm; matrix size, 384 × 269; NEX, 4), and proton density-weighted images with a 2-dimensional turbo spin echo sequence (TR, 2910 ms; TE, 23 ms; FOV, 12 × 12 cm; matrix size, 384 × 269; NEX, 4). RFC was defined by partial invisibility with surface irregularities on T1-weighted images after contrast enhancement and T2-weighted images. The images were reviewed blindly by two senior radiologists in a core laboratory. Cohen’s kappa was used to determine the intra-observer agreement in identification of the presence of RFC. The kappa value was 0.900 (*p* < 0.05).

### 2.3. Calculation of the CFD

CFD modeling and assessment of the hemodynamic features of the BAs were conducted centrally at GMU.

Three-dimensional BA. The three-dimensional geometry of the BA was reconstructed from TOF-MRA source images according to the previous report [11].

CFD procedure. The CFD procedure was run on the ANSYS software package version 19.0 according to Leng’s study [9]. A mesh was created on the vessel surface and within the vessel lumen in ANSYS ICEM CFD, with maximal element sizes of 0.1 for the inlets and outlets and 0.25 for the other parts of the mesh, containing more than 1 million elements in total for each case. Generic boundary conditions and blood properties were applied on the mesh in ANSYS CFX-Pre: pressures of 110 mmHg were applied at the inlets; the mass flow rates were estimated based on mean flow velocities from a population-based study applied at the outlets; a rigid arterial wall with a nonslip flow condition was assumed; and blood as an incompressible Newtonian fluid with a constant viscosity of 0.0035 kg m^−1^∙s^−1^ and density of 1060 kg m^−3^ was assumed. A blood flow simulation was conducted in ANSYS CFX by solving the Navier–Stokes equations; convergence was achieved when the root mean square residual value reached below 10^−4^.

The stenotic WSS was measured at the most severely narrowed cross-section. The proximal WSS was measured at the normal vessel segment proximal to the plaque (Figure 1). The WSS ratio was defined as follows: WSS ratio = Stenotic WSS/ proximal WSS.

The stenotic velocity was measured at the most severely narrowed cross-section. The proximal velocity was measured at the normal vessel segment proximal to the plaque. The velocity ratio was defined as follows: Velocity ratio = Stenotic velocity/proximal velocity.

The measurements were performed by two experienced neurologists. The first 20 patients were used to calculate Spearman’s rank correlation coefficient to assess the interobserver agreement between the two neurologists.

### 2.4. Statistics

The differences in the continuous variables were compared with the homogeneity test of variances. Student’s t-tests were used when the normality assumption was met; otherwise, the equivalent nonparametric test was used. The differences in the categorical variables were compared with Pearson’s Chi-square tests with post hoc analysis. A univariate binary logistic regression analysis was performed for the effect of the independent variables on the RFC. Individual variables with a *p* value < 0.1 in the univariate analysis were used in the multivariable regression analysis, and the results were expressed as odds ratios (ORs) and 95% confidence intervals (CIs). The area under the receiver operating characteristic (ROC) curve was used to assess the ability of the model to discriminate between patients with and without an RFC. *p* < 0.05 was considered statistically significant. SPSS 20.0 was used for the statistical analysis.

## 3. Results

From 1 June 2014, to 30 June 2019, 176 patients were enrolled in the present study (Figure 2). The clinical characteristics are summarized in Table 1. Spearman’s rank correlation coefficients of stenotic WSS, proximal WSS, stenotic velocity and proximal velocity were 0.997 (*p* < 0.05), 0.986 (*p* < 0.05), 0.994 (*p* < 0.05) and 0.982 (*p* < 0.05), respectively.

An RFC was identified in the 35 patients (19.9%, Figure 3). An RFC was significantly associated with acute infarction (27/35 vs. 96/141, *p* < 0.05, Table 1, Figure 1 and Figure 3).

The proximal WSS was higher in the patients with an RFC than in those without an RFC (8.68 ± 17.60 vs. 8.07 ± 4.55, *p* < 0.05, Table 1). The multivariable logistic regression analysis showed that proximal WSS was an independent risk factor for RFC (1.564, 95% CI 1.101–2.222, *p* < 0.05, Table 2).

The cut-off value of the proximal WSS was 4.84 Pa (area under the ROC curve 0.732, *p* = 0.008, 95% CI 0.565–0.899). In one patient with a lower proximal WSS, there was no infarction (Figure 3).

## 4. Discussion

In the present study, we used HR-MRI to identify basilar RFCs, which were significantly associated with stroke onset; a higher proximal WSS was an independent risk factor for a basilar RFC, and the cut-off value was 4.84 Pa. The strengths of our study were that we used the CFD to found the risk of basilar RFC, which was identified by HR-MRI.

HR-MRI is a noninvasive imaging modality to visualize the characteristics of plaques [12]. An RFC was the main characteristic of vulnerable plaque [13]. We used HR-MRI to identify RFCs in the BA plaques. RFCs occurred in one-fifth of the BA plaques with moderate to severe stenosis (50–99%). For other intracranial arteries, such as the MCA, RFCs was hard to visualize by 3.0 T HR-MRI [14] due to the different geometry. The diameter of the BA is larger than that of the MCA; the shape of the BA is a relative straight line [15]; and the BA has more cisternal space when compared with the MCA [16]. Therefore, BA plaques can be visualized more accurately on HR-MRI than MCA plaques. 

Next, we found that RFCs in the BA plaques were significantly associated with new stroke onset. It was partially supported by other studies. A recent study found that an RFC in the carotid artery indicated subsequent stroke onset [17]. The findings were confirmed by a meta-analysis that included 6210 subjects from 33 studies [18]. However, few studies have shown that an RFC of the intracranial artery predicts stroke onset in the same area [12]. This was the first report to provide evidence that an RFC in the intracranial artery was related to stroke onset.

Then, we explored the risk of RFC. Rupture was likely to occur when the mechanical stress exceeded the material strength of the fibrous cap [4]. The WSS is engaged in the development of plaque from atheroma to advantaged plaque [19]. The WSS is the interaction between blood flow and the plaque surface. Growing plaque changes the local arterial geometry, which results in changes in the magnitude, directionality, and spatial distribution of stress on the plaque [9]. It can not only cause fatigue damage to the fibrous cap, which would lead to the erosion of the plaque surface, but also trigger the sudden rupture of the fibrous cap [7]. In our study, increased WSS was associated with RFCs of BA plaques. Our result was consistent with others. Teng et al. found that stress on ruptured plaque was higher than that on unruptured plaque in the carotid artery [20]. An interesting thing was that we found the proximal WSS, not the stenotic WSS, was related to RFC. Jing et al. found that the maximum WSS was located in the rupture location of ruptured carotid plaques [2]. The maximum WSS was located in the stenotic throat, which was also proven by our findings. Rupture of vulnerable plaque is usually located in the shoulders of plaques [21], which might partially be due to the proximal WSS. The stress on the shoulder regions of plaques was responsible for 60% of plaque ruptures [22]. Previous studies tried transcranial doppler to measure the parameters of hemodynamics such as velocity intracranial artery [23,24]. However, vertebrobasilar artery had the lowest sensitivity for transcranial doppler [25] due to the tortuosity or asymmetry, which was common in the vertebrobasilar artery. Hence, CFD would be a better alternative.

Finally, we identified the threshold of the proximal WSS, which was 4.84 Pa. The true value of the WSS cannot be measured directly and can only be assessed by CFD calculation. The mean WSS of the normal carotid artery was 4.36 ± 1.32 Pa [26]. The mean WSS of carotid artery stenosis ranged from 30.9 ± 6.25 Pa to 212.65 ± 22.44 Pa [2]. For the coronary artery, a WSS greater than 2.5 Pa was associated with erosion of the fibrous cap [27]. The difference in the threshold might be due to the different geometry of the artery and plaque.

### Limitations

First, our study was a retrospective study. The limitations of retrospective studies also applied to our study. Second, some fibrous caps could not be visualized by HR-MRI. In the present study, we excluded these patients, which might have biased our results. For instance, intraplaque hemorrhage leads to a misdiagnosis of fibrous cap rupture in many patients who might have an RFC. In the coronary artery, investigators used optical coherence tomography or intravascular ultrasound to visualize the fibrous cap; however, these imaging tools were not widely used in the intracranial artery. Finally, it was ideal for identifying the risk for rupture-prone plaque.

## 5. Conclusions

In conclusion, our study demonstrated that the proximal WSS (>4.84 Pa) was an independent risk factor for basilar RFC, which was associated with acute infarction.

## Figures and Tables

**Figure 1 brainsci-12-01397-f001:**
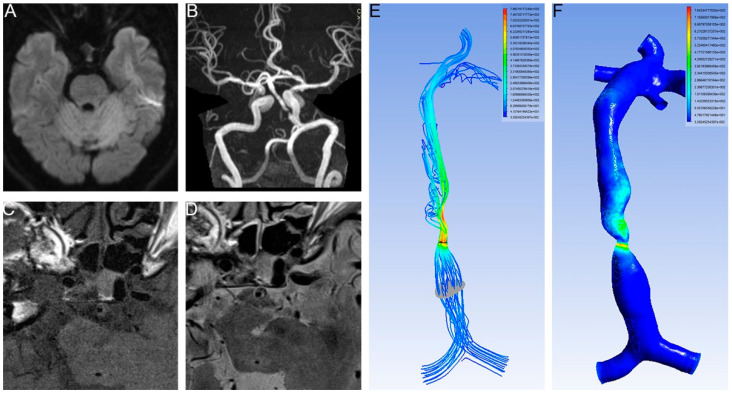
No infarction occurred in the lower proximal WSS. (**A**) DWI showed no infarction. (**B**) T2 images of HR MRI showed intact fibrous cap. (**C**) T1 images of HR MRI showed intact fibrous cap. (**D**) T2 images of HR MRI showed intact fibrous cap. (**E**) Velocity of BA. The planes (gray and dark) represent the place we measured. (**F**) WSS of BA. Proximal WSS was 2.88 Pa and stenotic WSS was 490.78 Pa.

**Figure 2 brainsci-12-01397-f002:**
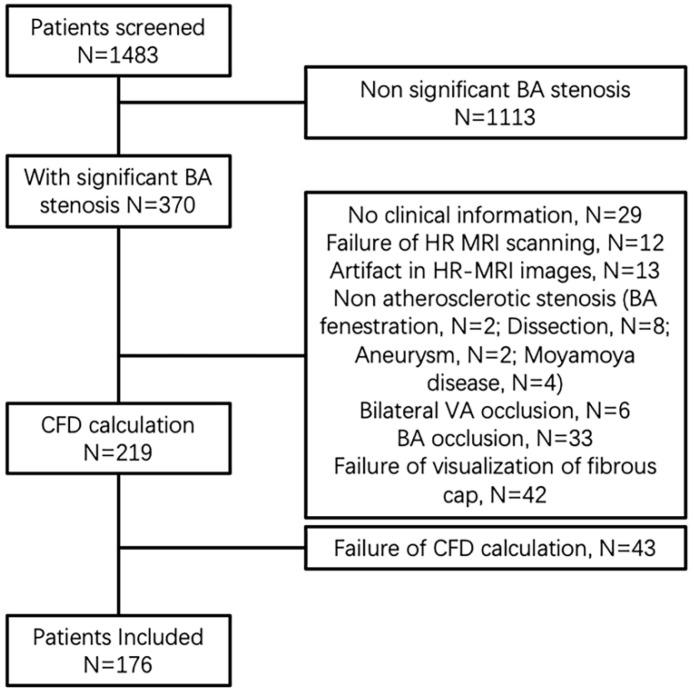
A flow diagram.

**Figure 3 brainsci-12-01397-f003:**
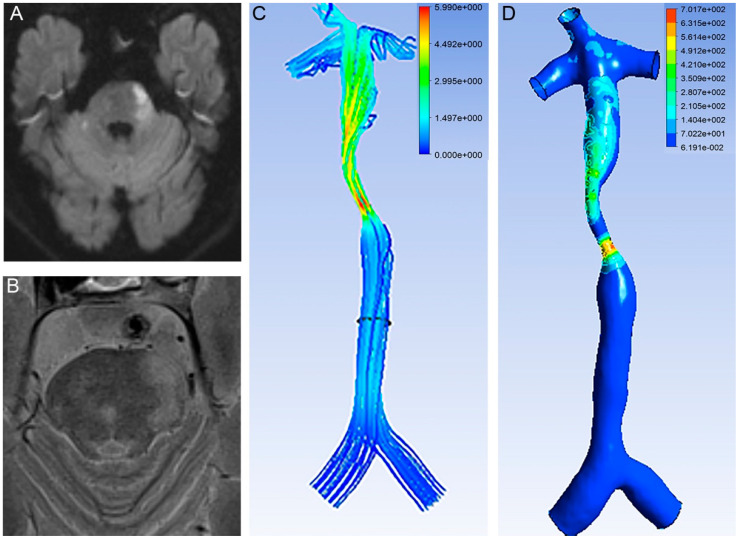
Higher proximal WSS related with stroke. (**A**) DWI showed an infarction located in the pontine. (**B**) T2 images of HR MRI showed RFC. (**C**) Velocity of BA. (**D**) WSS of BA. Proximal WSS was 7.34 Pa and stenotic WSS was 559.51 Pa.

**Table 1 brainsci-12-01397-t001:** The baseline characteristics.

Characteristics	Fibrous Cap		*p*
	Ruptured (*n*= 35)	Nonruptured (*n* = 141)	
Age (yrs)	53.3 ± 12.2	56.9 ± 12.9	0.156
Male	23	103	0.389
Medical History			
Hypertension	15	78	0.509
Diabetes	8	37	0.825
TC (mM)	3.89 ± 1.19	4.31 ± 1.19	0.095
LDL (mM)	2.83 ± 1.52	2.92 ± 0.97	0.682
TG (mM)	1.70 ± 0.64	1.98 ± 1.00	0.174
Smoker	9	54	0.387
Drinker	5	32	0.336
Medication			
Antihypertensive	12	67	0.239
Hypoglycemic	8	37	0.825
Statin	28	123	0.649
Antiplatelet	23	99	0.461
Previous stroke or TIA	1	24	0.049
New stroke onset	27	96	0.041
CFD			
Proximal WSS	8.68 ± 17.60	8.07 ± 4.55	0.014
Distal WSS	48.62 ± 147.26	103.78 ± 522.40	0.567
WSS ratio	9.88 ± 21.12	16.04 ± 70.63	0.557
Proximal velocity	0.34 ± 019	0.38 ± 0.12	0.275
Distal velocity	0.90 ± 1.31	1.05 ± 2.04	0.858
Velocity ratio	2.49 ± 2.85	2.91 ± 5.54	0.811

TG, triglyceride; TC, total cholesterol; LDL, low density lipoprotein; TIA, transient ischemic attack; CFD, Computational fluid dynamics; WSS, wall shear stress.

**Table 2 brainsci-12-01397-t002:** Multivariate logistic regression of the fibrous cap rupture.

Characteristics	B	S.E.	Wald	*p*	Exp(B)	95% CI	
TC	−0.621	0.463	1.797	0.18	0.538	0.217	1.332
New stroke	−0.177	0.754	0.055	0.815	0.838	0.191	3.672
Stroke history	−2.355	1.398	2.836	0.092	0.095	0.006	1.471
Proximal WSS	0.447	0.179	6.234	0.013	1.564	1.101	2.222
Distal WSS	−0.015	0.05	0.09	0.765	0.985	0.893	1.086
WSS ratio	0.252	0.153	2.721	0.099	1.287	0.954	1.737
Proximal velocity	−50.945	18	8.011	0.005	0	0	0
Distal velocity	12.52	5.383	5.41	0.02	273,843.797	7.168	1.0463 × 10^10^
Velocity ratio	−4.617	2.116	4.761	0.029	0.01	0	0.625
Proximal pressure	0	0.001	0.229	0.632	1	0.998	1.001
Distal pressure	−0.001	0.004	0.087	0.768	0.999	0.99	1.007
Pressure ratio	17.442	21.83	0.638	0.424	37,590,804.9	0	1.4357 × 10^26^

## Data Availability

Not applicable.

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
