# Peer review of "Increased Proximal Wall Shear Stress of Basilar Artery Plaques Associated with Ruptured Fibrous Cap"

_brainsci, 2022, doi:10.3390/brainsci12101397_

Round 1

Reviewer 1 Report

The paper presents the possibilities of assessing the risk of rupture of lesions in the basilar artery. As the authors noted, due to the lack of routine possibility of assessing intracranial arteries using techniques such as optical coherence tomography or intravascular ultrasound to visualize the fibrous cap, as well as methods of assessing the risk of basilar artery rupture.

According to the authors, the method presented in the article is ideal for identifying the risk for rupture-prone plaque.

The authors presented an alternative method to endovascular assessment such as OCT or IVUS. As the authors noted, both of these methods are widely used in interventional cardiology. Unfortunately, their use within intracranial arteries is limited, mainly due to tortuosity and difficulties with inserting these systems into intracranial segments, and additionally, it should be remembered that when trying to introduce an intracranial evaluation system, the examined artery may be damaged - and in the case of coronary arteries this is not a big problem, in the case of intracranial arteries it can cause very serious clinical complications.

This manuscript is relevant and interesting, because an additional assessment in the case of intracranial stenosis is very helpful in making a therapeutic decision. The subject is original and presents the subject in a different perspective than the previously presented works. The pressure at which plaque damage in the intracranial arteries may be damaged has been estimated - and this may be a useful parameter in qualifying a patient for procedure. Overall, the manuscript is well written, the text is clear and easy to read. The conclusions are consistent with the evidence and arguments presented, addressed the main question posed.

Author Response

We authors thank for this kind and positive comments. 

Reviewer 2 Report

 Huang et al are presenting an interesting study showing the association of wall shear stress of basilar artery plaques with ruptures fibrous cap. The scientific merit is indisputable, and the manuscript falls with the journal scope. I do have some comments to improve the manuscript:

1.       Although less sensitive in the posterior circulation, doppler ultrasound can improve the overall characterization of the dynamics of cervical-cranial circulation. The authors should discuss this.

2.       The images 78 were reviewed blindly by two senior radiologists in a core laboratory. Explain in what sense the radiologists were blinded?

3.       Were the neurologists doing the measurements of velocities blinded? if so, detail.

4.       Include bibliographic support to the information contained in Line 153-9 and 174-8 should

5.       “First, our study was a retrospective study. The limitations of retrospective studies 195 also applied to our study” – detail specifically in this study which limitations?

Author Response

  1. Although less sensitive in the posterior circulation, doppler ultrasound can improve the overall characterization of the dynamics of cervical-cranial circulation. The authors should discuss this.

Response: Thank you for your comment. As mentioned, ultrasound had less sensitivity to determine the dynamics in the posterior circulation. It was discussed in the revised manuscript.

  1. The images 78 were reviewed blindly by two senior radiologists in a core laboratory. Explain in what sense the radiologists were blinded?

Response: The initial reports of individual centers were blind to the radiologists in the core laboratory. Besides that, the two radiologists were blind to each other for the detection of RFC using HR-MRI images.

  1. Were the neurologists doing the measurements of velocities blinded? if so, detail.

Response: The two neurologists did the measurements of velocities using the CFD software independently. Spearman’s rank correlation coefficients was used to determine the intra-observer agreement of stenotic WSS, proximal WSS, stenotic velocity and proximal velocity as mentioned in the results section.

  1. Include bibliographic support to the information contained in Line 153-9 and 174-8 should

Response: The references were added in the revised manuscript.

  1. “First, our study was a retrospective study. The limitations of retrospective studies 195 also applied to our study” – detail specifically in this study which limitations?

Response: Retrospective might introduce selection bias for some subjects missed the key information. For instance, we excluded 29 patients because of no clinical information. Secondly, it was hardly to control exposure assessment. We divided the patients into two groups: rupture and non-rupture. There was limited number in the rupture group while some patients were excluded due to low quality of MRI images. Finally, some key statistics cannot be measured. No follow up was performed in this retrospective study.